# Study on the Combustion Mechanism of Diesel/Hydrogen Dual Fuel and the Influence of Pilot Injection and Main Injection

Longlong Xu, Haochuan Dong, Shaohua Liu *, Lizhong Shen and Yuhua Bi

Yunnan Province Key Laboratory of Internal Combustion Engines, Kunming University of Science and Technology, Kunming 650504, China; xulonglong@stu.kust.edu.cn (L.X.); haochuan97@gmail.com (H.D.); lzshen@foxmail.com (L.S.); yuhuabi97@sina.com (Y.B.)
* Correspondence: liushaohua@kust.edu.cn; Tel.: +86-136-0880-1893

**Abstract:** Hydrogen is a clean and renewable alternative fuel. In this paper, the combustion mechanism of diesel/hydrogen dual fuel is constructed and verified. The mechanism is combined with three-dimensional numerical simulation to study the effects of pilot injection and main injection on the combustion and emissions of a diesel/hydrogen dual fuel engine. The mechanism uses a 70% mole fraction of n-decane and 30% mole fraction of $\alpha$-methylnaphthalene as diesel substitutes, and it combines n-decane, $\alpha$-methylnaphthalene, $NO_X$, PAH, soot and $H_2/C_1$-$C_3$ sub-mechanisms to form a diesel/hydrogen dual fuel combustion mechanism. The mechanism was verified by chemical kinetics, including the ignition delay time, JSR (Jet Stirred Reactor) oxidation and laminar flame speed, and then, it was verified by computational fluid dynamics. The results show that the simulated values are in good agreement with the experimental values of cylinder pressure, heat release rate and emissions data. The mechanism can well predict the combustion and emissions of a diesel/hydrogen dual fuel engine. Compared with single injection, the peak heat release rate, peak cylinder pressure and MPIR (Maximum Pressure Rise Rate) increase with the increase in pilot mass percent from 5% to 20%, which makes the phase of CA10 and CA50 advance and reduces CO emissions, but $NO_X$ emissions increase. With the advance of pilot injection timing from 10° CA BTDC to 30° CA BTDC, the peak cylinder pressure increases, the peak heat release rate decreases, CA10 and CA50 advance, CO emissions decrease, $NO_X$ emissions increase and $NO_X$ emissions peak at 30° CA BTDC. When the pilot injection timing is further advanced from 30° CA BTDC to 50° CA BTDC, the peak cylinder pressure decreases, the peak heat release rate increases, CA10 and CA50 are delayed, CO and $NO_X$ emissions are reduced, and $NO_X$ emissions at 50° CA BTDC are lower than those at 10° CA BTDC. With the advance of main injection timing from 0° CA BTDC to 8° CA BTDC, CO emissions decrease, $NO_X$ emissions increase, the peak cylinder pressure increases, the peak heat release rate decreases slightly first and then increases, and the peak cylinder pressure and peak heat release rate corresponding to the overall phase shift forward. When the main injection timing is advanced to 6° CA BTDC, MPIR is 1.3 MPa/° CA, exceeding the MPIR limit of diesel engine 1.2 MPa/° CA.

**Keywords:** diesel/hydrogen; dual fuel combustion mechanism; pilot injection; main injection; combustion and emissions



## 1. Introduction

Fossil fuels such as diesel and gasoline are the most widely used fuels in the field of transportation [1]. Compressed ignition (CI) engines with diesel as fuel have the advantages of high energy efficiency and high torque, so they play a vital role in heavy transportation, agriculture, power generation and other fields [2–4]. However, the emissions released by diesel engine combustion will further aggravate environmental degradation and global warming. Among them, $CO_2$ emissions will lead to enhanced greenhouse effect, and $NO_X$ and PM emissions will have adverse effects on human health [5,6]. At present, China is the world 's largest carbon emitter. To reduce carbon emissions, China has committed to

achieving a carbon peak by 2030 and achieving carbon neutrality by 2060 [7–9]. Energy conservation and emissions reduction is the common goal of the world. It is of great significance to develop low-carbon emissions engines and explore clean energy sources such as hydrogen and ammonia to replace diesel [10]. Among various alternative fuels, hydrogen has the advantages of wider flammability limit, higher laminar flame speed, higher calorific value, and lower gas ignition energy, [11] and hydrogen combustion does not produce $CO_2$ emissions, either. It is recognized as a clean and renewable energy source, and many countries have introduced policies to support the development of hydrogen energy [12–17].

Due to the high compression ratio and the equivalence ratio of hydrogen–air mixture being greater than 1, the engine with pure hydrogen as fuel has the disadvantage of having a knock phenomenon [18–20]. When hydrogen is combined with diesel in a CI engine, it has the advantages of low carbon emissions, high operability and good adjustable performance while ensuring high combustion efficiency. Therefore, in recent years, the engine with hydrogen as the main fuel and diesel to ignite hydrogen has attracted much attention. Qin et al. [21] carried out experiments on engine performance under different hydrogen replacement rates. The results showed that when the hydrogen replacement rate was 20%, the in-cylinder pressure increased by 7.7%. Jianbin Luo et al. [22] found that with the addition of hydrogen, the heat release rate of the diesel engine increased, the flame propagation speed accelerated, the combustion duration shortened, and the pressure and temperature in the cylinder increased. Juknelevicius et al. [23] carried out experimental research on hydrogen blending in diesel engines. After hydrogen blending, the ignition delay period of diesel engines is shortened, and $NO_X$ emissions are reduced. When the hydrogen flow rate increases to 30 L/min, the thermal efficiency of the engine increases, while the specific fuel consumption is lower than that of the pure diesel mode. Szwaja and Grab-Rogalinski [24] carried out pure hydrogen and diesel/hydrogen dual fuel operations on a diesel engine, respectively. It was found that the combustion of diesel/hydrogen dual fuel was milder than that of pure hydrogen mode, and no knock occurred when the hydrogen replacement rate was low. Rorimpandey et al. [25] used a high-speed camera and pressure-sensing equipment to study the combustion phenomenon of the direct injection of hydrogen and n-heptane in a constant volume combustion chamber. The results show that the hydrogen beam needs to be mixed with the n-heptane liquid beam to a certain extent before it can be ignited, the heat release rate in this direct injection mode is mainly affected by hydrogen combustion, and the ignition delay time is longer. Suzuki et al. [26] studied the performance and emissions of a diesel/hydrogen dual fuel engine on an in-line four-cylinder engine. Experiments showed that with the increase in hydrogen replacement rate, the emissions of $CO_2$, HC and soot decreased, the thermal efficiency of the engine increased, and $NO_X$ emissions increased. Jabbr et al. [27] confirmed that in the diesel/hydrogen dual fuel mode, $NO_X$ and soot emissions show a trade-off relationship with the increase in hydrogen replacement rate.

However, it is expensive and time-consuming to intensively study the combustion and emissions characteristics of dual fuel engines only through experiments. For the ignition diesel/hydrogen dual fuel engine, it is a convenient and efficient method to combine the detailed chemical reaction mechanism with computational fluid dynamics (CFD) to carry out the three-dimensional simulation of engine combustion and emissions. With the development of alternative energy sources and new combustion concepts, the development of a dual fuel combustion mechanism has become one of the research hotspots in recent years.

Xu et al. [28] constructed an ammonia-n-heptane mechanism containing 69 components and 389 elementary reactions by combining decoupling and optimization methods. Guo et al. [29] used n-hexadecane to characterize diesel; combining a direct relationship diagram method, error-based direct relationship diagram method, sensitivity analysis and reaction path analysis, they optimized the reduced diesel engine mechanism, combined the natural gas mechanism for secondary simplification to obtain the natural gas–diesel

engine mechanism of 155 components and 645 elementary reactions, and applied it to the CFD simulation of internal combustion engines. The mechanism is verified to be accurate, and the three-dimensional combustion state of a natural gas–diesel RCCI engine can be simulated by a numerical model. Schuh et al. [30] constructed a methane–propane–n-heptane mechanism and studied the similarities and differences of ignition delay time of several different n-heptane mechanisms in a simulated natural gas–diesel combustion, but the mechanism construction still needs a more detailed treatment of the reaction rate parameters. Wang et al. [31] constructed an n-heptane–n-butanol–PAH mechanism with 78 components and 342 elementary reactions and carried out engine CFD simulation on this mechanism. The results show that the mechanism calculation results are in good agreement with the shock tube, constant volume combustion bomb and engine operation data, and they can effectively provide strong support for how to achieve the best combustion and emissions coordination in the diesel–n-butanol dual fuel mode. The numerical simulations of dual fuel engines reviewed above reveal that a smaller-scale, more accurate and reliable diesel/hydrogen dual fuel chemical kinetic mechanism is still not available, and research in this aspect is greatly needed.

The analysis of dual fuel engines above shows that researchers need a more suitable diesel/hydrogen dual fuel chemical kinetic mechanism to predict the combustion and emissions of dual fuel engines directly. In this paper, a 70% mole fraction of n-decane and 30% mole fraction of $\alpha$-methylnaphthalene were selected as diesel substitutes to construct and verify the reduced chemical kinetic mechanism of diesel/hydrogen dual fuel. The reduced mechanism was applied to a high-pressure common rail diesel engine for three-dimensional simulation research, and the effects of pilot injection and main injection on the combustion and emissions of diesel/hydrogen dual fuel engine were investigated. The results show that the mechanism has a good prediction effect, which can provide reference for the research of diesel/hydrogen dual fuel engines.

## 2. Mechanism Development

This section mainly describes the construction of the diesel/hydrogen dual fuel combustion mechanism, including the n-decane sub-mechanism, $\alpha$-methylnaphthalene sub-mechanism, $H_2/C_1$-$C_3$ sub-mechanism, PAH sub-mechanism, soot sub-mechanism, and $NO_X$ sub-mechanism. The ignition delay sensitivity analysis of the combined diesel/hydrogen dual fuel combustion mechanism was carried out, and the pre-exponential factor of some elementary reactions was adjusted to optimize the original mechanism. Finally, the diesel/hydrogen dual fuel combustion mechanism with 191 components and 847 elementary reactions was obtained.

### 2.1. The Reduce Mechanism of N-Decane and A-Methylnaphthalene

Choosing a 70% mole fraction of n-decane and 30% mole fraction of $\alpha$-methylnaphthalene (IDEA reference fuel) as diesel substitutes, it has been proved to be close to real diesel in fuel injection and atomization, spontaneous combustion, flame propagation, turbulence field, pollutant formation and so on [32–34]. The main physical and chemical properties of the IDEA reference fuel and real diesel are shown in Table 1. According to the oxidation path of straight-chain alkanes at low/high temperatures proposed by Chang et al. [35], the n-decane mechanism was constructed. The reaction pathway can describe the oxidation of n-octane to n-hexadecane. The reaction pathways of n-decane at pressure of 10.0 atm, equivalence ratio of 1.0 and temperatures of 750 (low temperature) and 1150 That's correctK (high temperature) are shown in Figure 1. The $\alpha$-methylnaphthalene mechanism consists of two parts, which are selected from the $\alpha$-methylnaphthalene oxidation mechanism of Mati et al. [36] and the toluene mechanism of Wang Hu et al. [37]. Figure 2 shows the main reaction path of the $\alpha$-methylnaphthalene mechanism under the conditions of temperature of 1150 K, pressure of 10.0 atm and equivalence ratio of 1.0.

**Table 1.** Major physiochemical properties of IDEA reference fuel and diesel.

| Type of Fuel | Cetane Number | Low Heating Value | Density at 20 °C (kg/m³) | H/C |
|---|---|---|---|---|
| IDEA fuel | 56 | 42.4 | 817 | 1.7 |
| Diesel | 53 | 42.8 | 840 | 1.76 |

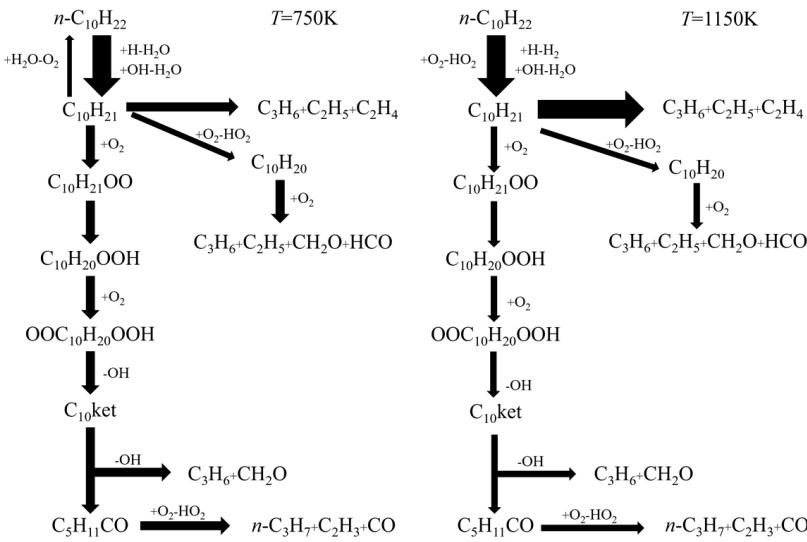

**Figure 1.** Reaction pathway diagram of n-decane sub-mechanism.

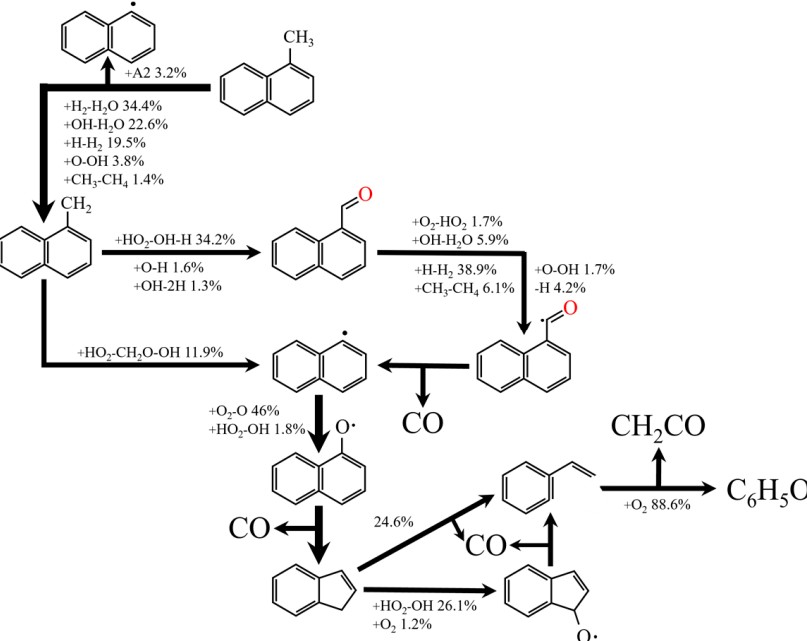

**Figure 2.** Reaction pathway diagram of α-methylnaphthalene sub-mechanism.

### 2.2. Construction of Diesel/Hydrogen Dual Fuel Mechanism

The reduced mechanism of n-decane and α-methylnaphthalene needs to be coupled with $NO_X$, PAH, soot and $H_2/C_1$-$C_3$ sub-mechanisms. The $H_2/C_1$-$C_3$ sub-mechanism is selected from the $C_1$-$C_3$ mechanism of Wang et al. [38]. The PAH sub-mechanism is derived from the detailed mechanism of Slavinskaya et al. [39], which is used to predict the formation of PAH in ethylene–ethane flames. The soot sub-mechanism is derived from Frenklach [40]. The specific diesel/hydrogen dual fuel mechanism construction framework is shown in Figure 3. The mechanism of n-decane and α-methylnaphthalene describes the process of cracking and oxidizing n-decane and α-methylnaphthalene into smaller molecules,

respectively, and then, it is coupled with the $H_2/C_1$-$C_3$ sub-mechanism to describe the further oxidation of such small molecules into $CO_2$ and $H_2O$. The PAH sub-mechanism describes the process of generating soot precursors from small molecules. The $NO_X$ sub-mechanism is used to predict the formation of nitrogen oxides. The diesel/hydrogen dual fuel mechanism construction process is shown in Figure 4.

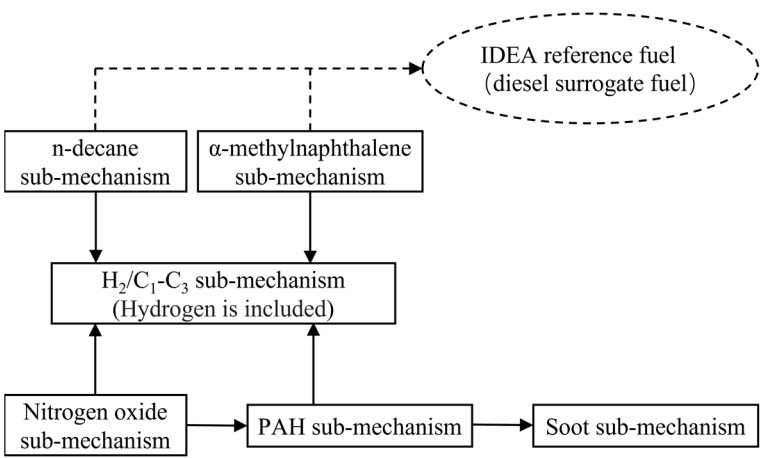

**Figure 3.** Schematic diagram of the hydrogen/diesel combustion mechanism.

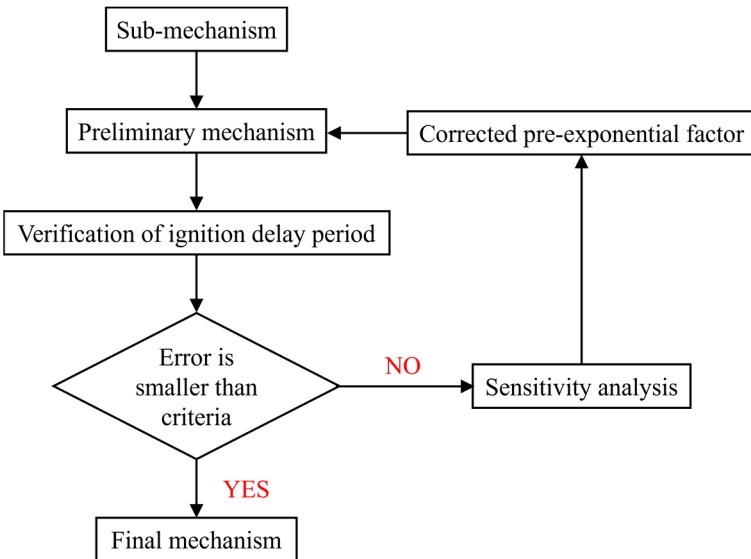

**Figure 4.** Construction process of the diesel/hydrogen combustion mechanism.

The above sub-mechanism is coupled to obtain the diesel/hydrogen dual fuel mechanism, and the ignition delay time of the mechanism is preliminarily verified. It is found that the simulated value of the hydrogen ignition delay time has a large error with the experimental value. Therefore, it is necessary to analyze and adjust it to optimize the coupling mechanism. To determine the key reaction of ignition delay, the sensitivity analysis of ignition delay of the hydrogen and n-decane/α-methylnaphthalene mixture was carried out, as shown in Figures 5 and 6. The normalized temperature sensitivity coefficient is positive, which indicates that it promotes ignition. Conversely, a negative normalized sensitivity coefficient indicates the suppression of ignition. The ignition delay sensitivity analysis of hydrogen is shown in Figure 5. At 1000 K, R565, R580 and R582 are most sensitive to the ignition delay of hydrogen. Finally, the modified Arrhenius Formula (1) is used to optimize the ignition delay reaction. The adjustment results are shown in Table 2.

$$k = AT^\beta e^{-\frac{E_a}{RT}} \tag{1}$$

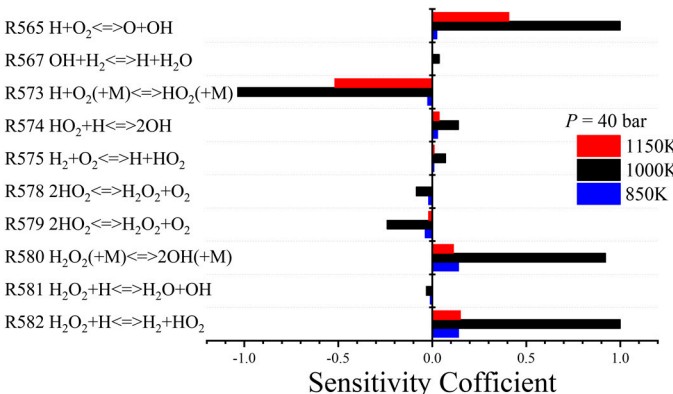

**Figure 5.** Sensitivity analysis of ignition delay times for the hydrogen/air mixtures.

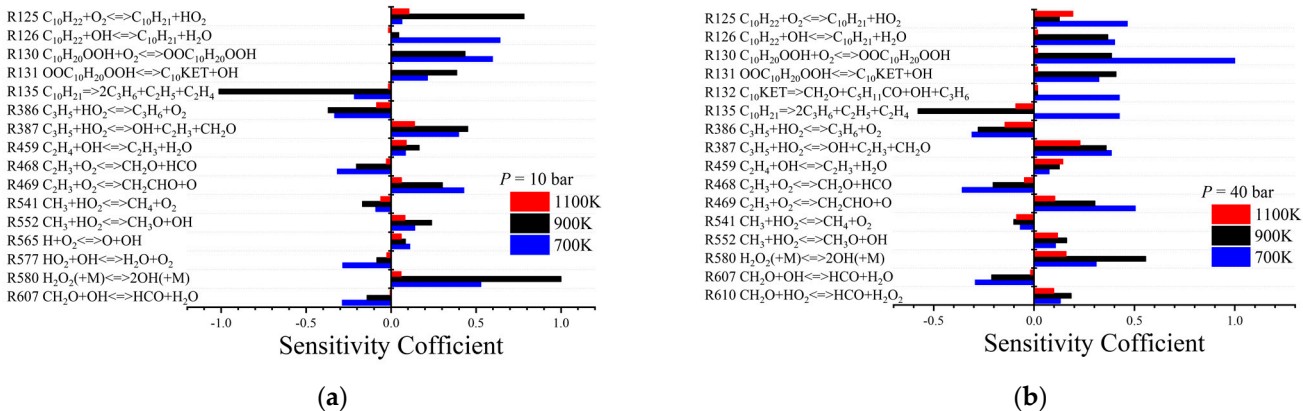

(**a**)                                         (**b**)

**Figure 6.** Sensitivity analysis of ignition delay times for the n-decane-$\alpha$-methylnaphthalene blends: (**a**) P = 10 bar; (**b**) P = 40 bar.

**Table 2.** Adjustment of the pre-exponential factors.

| Elementary Reaction | | A | |
|---|---|---|---|
| Reaction Number | Reaction Chemical Equation | Before Adjustment | After Adjustment |
| R565 | $H + O_2 \Leftrightarrow O + OH$ | $9.33 \times 10^{13}$ | $1.90 \times 10^{14}$ |
| R573 | $OH + H_2 \Leftrightarrow H + H_2O$ | $1.17 \times 10^9$ | $2.20 \times 10^8$ |
| R580 | $H_2O_2(+M) \Leftrightarrow 2OH(+M)$ | $1.30 \times 10^{17}$ | $1.20 \times 10^{17}$ |
| R582 | $H_2O_2 + H \Leftrightarrow H_2 + HO_2$ | $1.60 \times 10^{12}$ | $6.02 \times 10^{13}$ |

In the formula, $A$ is the pre-exponential factor ($cm^3 \cdot mol^{-1} \cdot s^{-1}$); $T$ is the thermodynamic absolute temperature (K); $\beta$ is a temperature index, which is a dimensionless number; $Ea$ is the activation energy ($cal \cdot mol^{-1}$); and $R$ is the ideal gas constant, $R = 8.31446261815324 \ J \cdot mol^{-1} \cdot K^{-1}$.

After adjustment, the modified diesel/hydrogen dual fuel mechanism was used to analyze the sensitivity of the IDEA reference fuel to ignition delay under the conditions of 700 K (low temperature), 900 K (NTC) and 1100 K (high temperature), 10 bar and 40 bar, and the equivalence ratio of 1.0. The most sensitive reactions under various temperature conditions are shown in Figure 6. In Figure 6, the reactions that have a greater impact on the ignition of the IDEA reference fuel can be divided into three categories: the initial oxidation reaction of n-decane (R125-R135 in Figure 6a,b), the oxidation reaction of small molecule $C_1$-$C_3$ (R386-R552 and R607 in Figure 6a, R386-R552 and R607-R610 in Figure 6b), and the reaction of $H_2/O_2$ system (R565-R580 in Figure 6a, R580 in Figure 6b). The final diesel/hydrogen dual fuel mechanism includes 191 components and 847 elementary reactions.

## 3. Mechanism Validation

### 3.1. Validation of Ignition Delays

Ignition delay time is the period from the critical ignition condition to the flame of the combustible mixture. It is a very important parameter, which determines the mixing time of the fuel and affects the distribution, combustion, and emissions characteristics of the fuel. In Figure 7a,b, the experimental values are derived from the α-methylnaphthalene–n-decane–air blends ignition operation carried out by Wang et al. [41] in the shock tube. The experimental values in Figure 7c,d are derived from the ignition delay data of DF-2 diesel measured by Alturaifi et al. [42], F-76 and Shell GTL diesel measured by Gowdagiri et al. [43], and DF-2 and Europe DF-2 diesel measured by Haylett et al. [44]. The experimental values in Figure 7e,f are derived from the shock tube experiments of Herzler and Naumann [45]. It can be seen from Figure 7 that the error between the simulated and experimental values of the ignition delay time of the n-decane-α-methylnaphthalene blends, IDEA reference fuel and hydrogen is small, and the average error is within one order of magnitude. Therefore, the diesel/hydrogen dual fuel mechanism can reasonably predict the ignition delay time of n-decane–α-methylnaphthalene blends, IDEA reference fuel and hydrogen. The IDEA reference fuel as a diesel alternative is reasonable and can better reflect the ignition delay time of diesel.

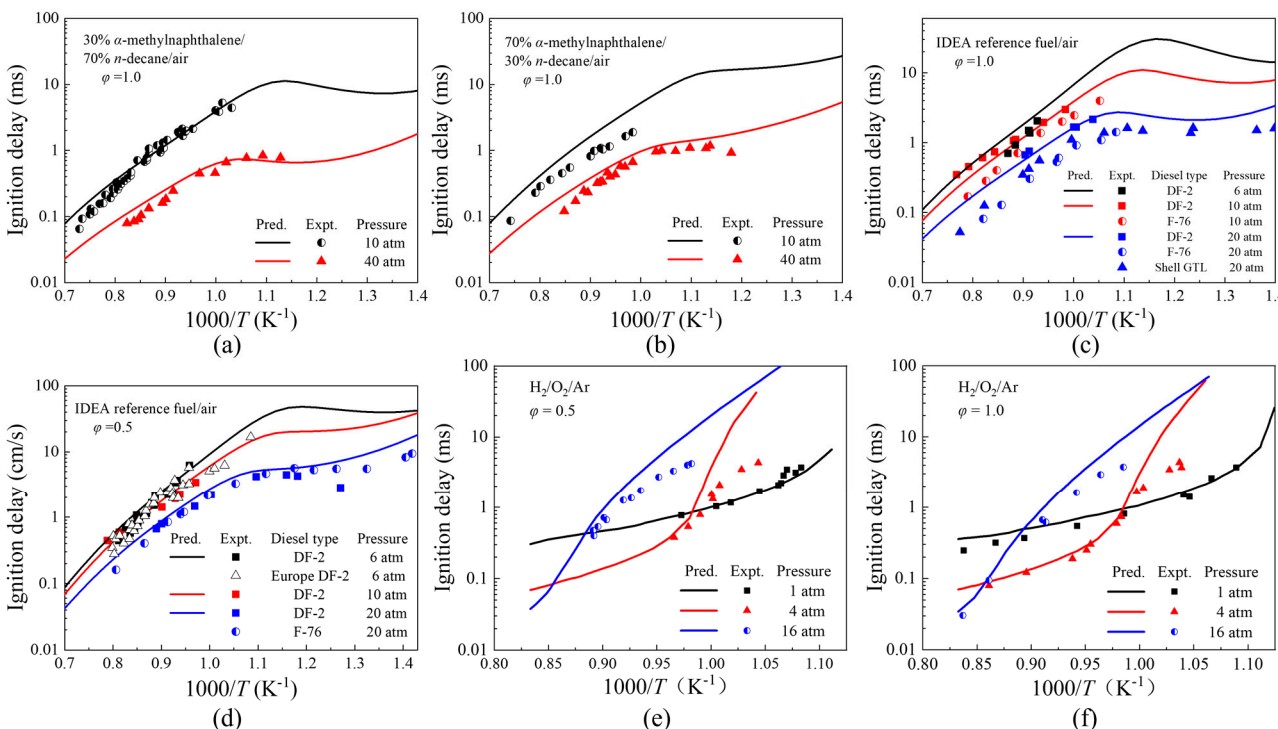

**Figure 7.** Comparison of the experimental and predicted ignition delays for α-methylnaphthalene/n-decane mixtures, IDEA reference fuel and hydrogen: (**a**) 30% α-methylnaphthalene/70% n-decane/air, $\varphi$ = 1.0; (**b**) 70% α-methylnaphthalene/30% n-decane/air, $\varphi$ = 1.0; (**c**) IDEA reference fuel/air, $\varphi$ = 1.0; (**d**) IDEA reference fuel/air, $\varphi$ = 0.5; (**e**) $H_2/O_2/Ar$, $\varphi$ = 0.5; (**f**) $H_2/O_2/Ar$, $\varphi$ = 1.0.

### 3.2. Species Concentration Profile

The essence of combustion is a violent oxidation reaction. The JSR (Jet Stirred Reactor) is an ideal continuous stirred tank reactor, which is suitable for the study of fuel oxidation and cracking [46]. Figure 8a–d show the concentration of key substances in IDEA reference fuel and diesel JSR oxidation operation [47]. The experimental value of Figure 8e,f is the hydrogen oxidation operation carried out by Cong et al. in a JSR [48]. It can be seen from the information in the figure that the simulated values of the concentration of key components of IDEA reference fuel and hydrogen have small errors with the experimental values, and

the trend with temperature changes is more consistent. Therefore, the diesel/hydrogen dual fuel mechanism can reasonably predict the oxidation process of diesel and hydrogen.

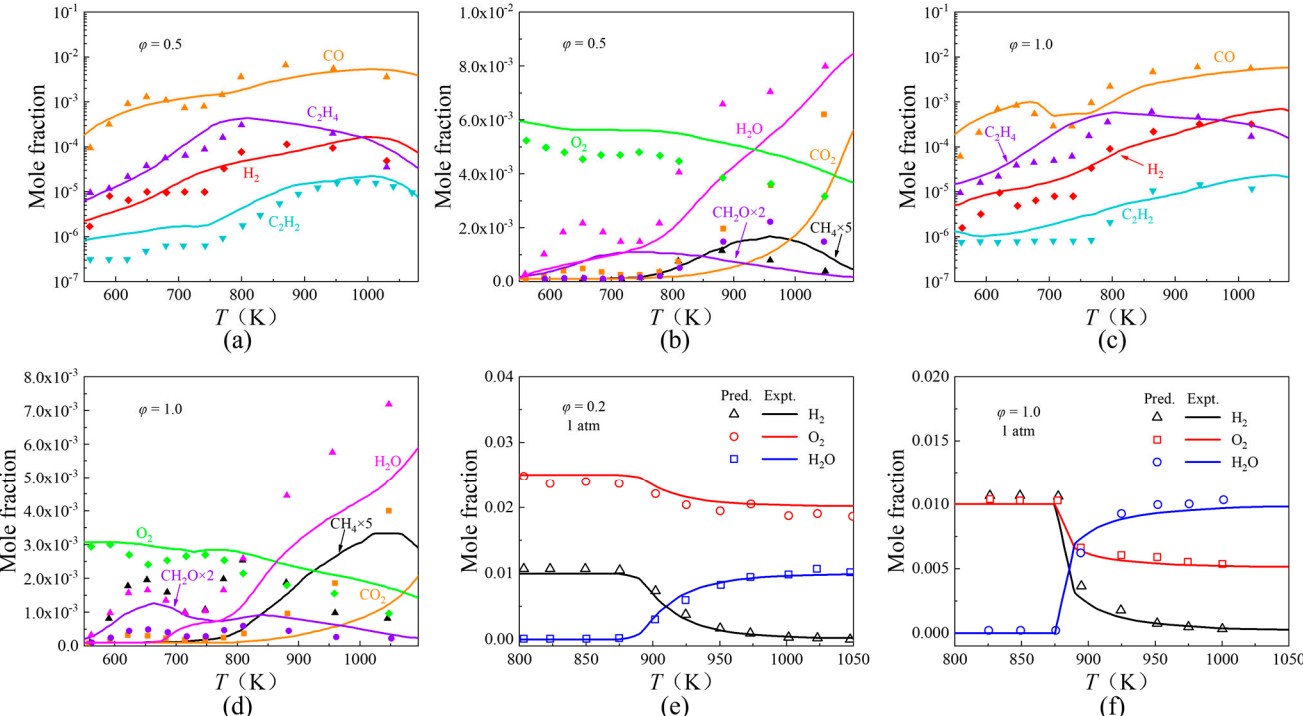

**Figure 8.** Comparison of the experimental and predicted species concentration profiles for IDEA reference fuel and hydrogen: (**a**,**b**) IDEA reference fuel, $\varphi$ = 0.5; (**c**,**d**) IDEA reference fuel, $\varphi$ = 1.0; (**e**) $H_2$, $\varphi$ = 0.2; (**f**) $H_2$, $\varphi$ = 0.5.

## 3.3. Laminar Flame Speeds

Laminar flame propagation velocity is an important parameter to reflect the combustion characteristics, which can characterize whether the premixed combustion process is intense [49,50]. Figure 9 shows the comparison between the simulated and experimental values of laminar flame velocities of n-decane, α-methylnaphthalene, IDEA reference fuel and hydrogen. Among them, the experimental value of n-decane comes from [51], the experimental value of α-methylnaphthalene comes from [52], the experimental value of diesel comes from [53,54], and the experimental value of hydrogen comes from [55,56]. It can be seen from the information in the figure that under different temperature conditions, the simulated values of laminar flame velocities of different fuels are consistent with the experimental values with the change of equivalence ratio, and the average error does not exceed 16%. Therefore, the diesel /hydrogen combustion mechanism can reasonably predict the laminar flame velocities of diesel substitutes, diesel, and hydrogen.

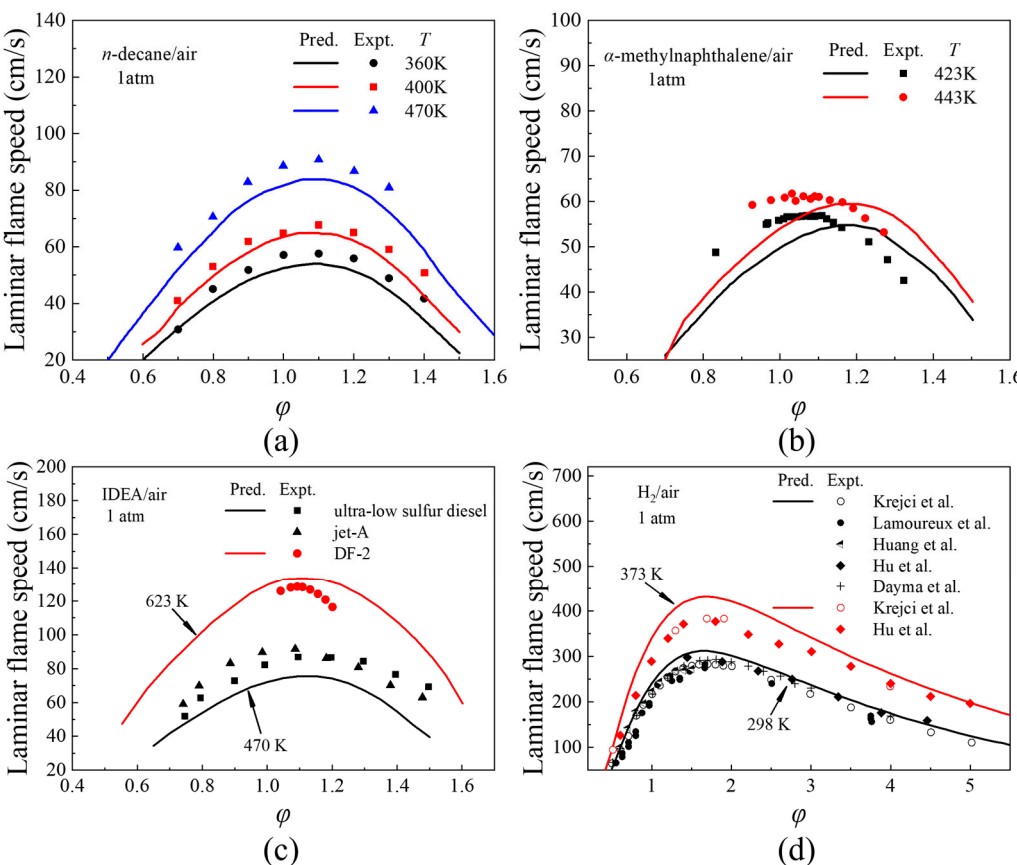

**Figure 9.** Comparison of the experimental and predicted ignition delays for n−decane, α-methylnaphthalene, IDEA reference fuel and hydrogen: (**a**) n-decane/air, P = 1 atm, T = 360 K, 400 K, 470 K; (**b**) α-methylnaphthalene/air, P = 1 atm, T = 423 K, 443 K; (**c**) IDEA/air, P = 1 atm, T = 470 K, 623 K; (**d**) $H_2$/air, P = 1 atm, T = 298 K, 373 K.

## 4. CFD Model Construction and Validation

In this section, the three-dimensional CFD model is constructed and verified according to the combustion mechanism of diesel/hydrogen dual fuel proposed above. Under the condition of verifying the validity of the model, the operation schemes of different pilot injection and main injection are formulated to study the effects of pilot injection quantity, pilot injection timing and main injection timing on the combustion and emissions of diesel/hydrogen dual fuel engine.

### 4.1. CFD Model Construction

Ansys Forte is used to study the three-dimensional simulation model. The basic control equations of ANSYS Forte include mass conservation equation, fluid continuity equation, kinetic energy conservation equation, energy conservation equation and gas phase mixture state equation. The physical model used in the study is shown in Table 3. The RNG k-ε model is used to simulate turbulence (the Reynolds number (Re) is set to 130), the O'Rourke–Amsden model is used to simulate the droplet collision process, the KH-RT model is used to describe the droplet breakup process, the ROI (Radius of Influence) model is used to simulate the droplet collision process, the discrete multi-component fuel evaporation model (DMC) is used to simulate the spray droplet evaporation process, and the Han–Reitz model is used to simulate the wall heat transfer process.

**Table 3.** Computational sub-models for the CFD simulation.

| Model Type | Name |
|---|---|
| Turbulence model | RNG k-ε |
| Droplet collision model | Radius of Influence |
| Spray crushing model | KH-RT |
| Wall oil film model | O'Rourke–Amsden |
| Evaporation model | DMC |
| Wall heat transfer model | Han–Reitz |

The hydrogen replacement rate Is calculated by Formula (2), as shown below:

$$R_{H_2} = \frac{m_{H_2} LHV_{H_2}}{m_{H_2} LHV_{H_2} + m_d LHV_d} \times 100\%, \tag{2}$$

In the formula, $m_{H_2}$ is the hydrogen flow rate, and the unit is kg·h$^{-1}$; $LHV_{H_2}$ is the low calorific value of hydrogen, and the unit is MJ·kg$^{-1}$; $m_d$ is the diesel flow rate, and the unit is kg·h$^{-1}$; and $LHV_d$ is the low calorific value of diesel, and the unit is MJ·kg$^{-1}$.

A diesel/hydrogen dual fuel engine operation was carried out on an in-line four-cylinder high-pressure common rail diesel engine. Hydrogen is injected from the hydrogen rail into the intake port for multi-point injection and mixed with air into the cylinder. The diesel in the high-pressure common rail pipeline ignites the hydrogen–air mixture, thereby realizing the operation of the diesel/hydrogen dual fuel engine. Table 4 is the engine specification. Figure 10 shows the schematic diagram of engine bench installation.

**Table 4.** Engine parameters of YN D30TCIF.

| Parameter | Value |
|---|---|
| Engine type | 4 cylinders in line, high-pressure common rail, supercharged medium cold-pressure combustion engine |
| Rated speed (r/min) | 3200 |
| Rated power (kW) | 115 |
| Maximum torque (N·m) | 450 |
| speed at maximum torque (r/min) | 1800 |
| Compression ratio | 16.6:1 |
| Bore×Stroke (mm) | 95 × 105 |
| Displacement (L) | 2.977 |
| Oil atomizer | Bosch, 8-nozzle |

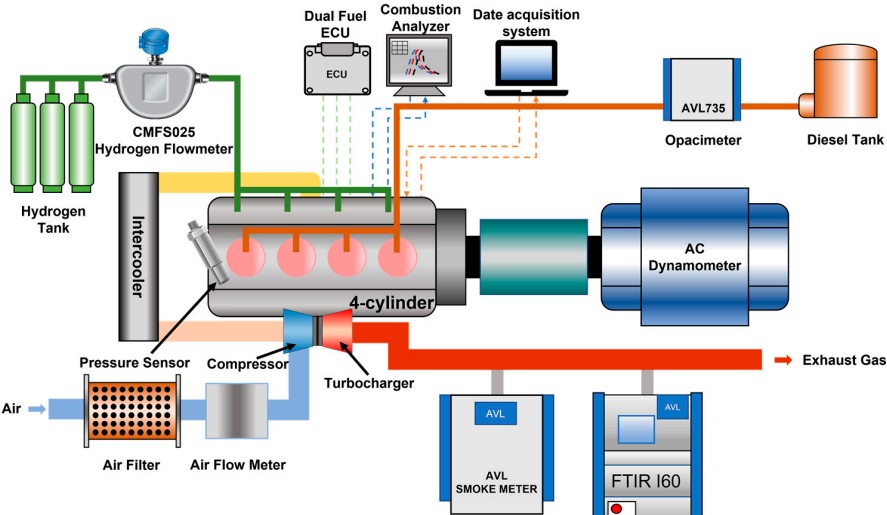

**Figure 10.** Diagram of operation bench.

The simulation model is based on the actual shape and size of the engine: using the symmetry between the center of the injector cylinder and the nozzle hole and the shape of the nozzle hole of the injector, to save the calculation time, a 1/8 three-dimensional calculation model is established, as shown in Figure 11. The intake valve closing (IVC) time is set to −125° CA, the exhaust valve opening (EVO) time is 130° CA, and the upper dead point corresponds to the crankshaft angle of 0° CA. The valve timing selected by the three-dimensional model in this paper is consistent with the engine valve timing during the operation. The initial swirl ratio is assumed to be 1.6, the spray angle is 120°, and the spray cone angle is 12°. Table 5 gives the initial conditions and boundary conditions.

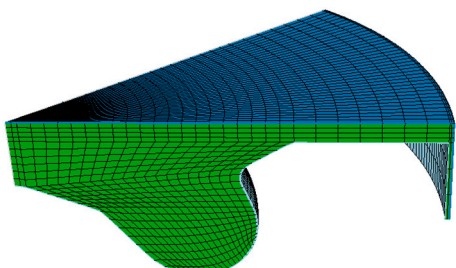

**Figure 11.** Computational mesh at top dead center.

**Table 5.** Initial conditions and boundary conditions.

| Parameter | Value |
| --- | --- |
| Intake pressure (bar) | 2.9 |
| Intake temperature (K) | 390 |
| Turbulent kinetic energy ($m^2/s^2$) | 21.6 |
| Turbulent length scale (cm) | 0.483 |
| Swirl ratio | 1.6 |
| Spray angle | 120° |
| Spray cone angle | 12° |
| Nozzle hole number | 8 |
| Nozzle hole diameter (mm) | 0.259 |
| IVC (CA BTDC) | −125° |
| EVO (CA BTDC) | 130° |
| Cylinder head | Wall, 525 K |
| Piston | Mesh movement, 525 K |
| Liner | Wall, 400 K |

## 4.2. CFD Model Validation

To verify the prediction accuracy of the dual fuel combustion mechanism and the three-dimensional simulation model, based on the bench operation data of the diesel/hydrogen dual fuel engine, four operation conditions with a speed of 1800 r/min and a torque of 322 N·m and 365 N·m were selected to verify the combustion and emissions characteristics of the diesel/hydrogen dual fuel engine. The operating parameters are shown in Table 6.

**Table 6.** Engine operation conditions for the CFD validation.

| Parameters of Operation | Operation Condition | | | |
| --- | --- | --- | --- | --- |
| | a | b | c | d |
| Engine speed (r/min) | 1800 | 1800 | 1800 | 1800 |
| Torque/(N·m) | 322 | 322 | 365 | 365 |
| Hydrogen replacement rate/% | 0 | 30 | 0 | 27.01 |
| Engine fuel consumption/(kg·h$^{-1}$) | 12.98 | 8.97 | 14.34 | 10.35 |

Figure 12a–d show the comparison chart of the simulated and experimental values of the cylinder pressure and heat release rate of the engine under four experimental conditions. It can be seen from Figure 12 that the cylinder pressure and heat release rate curves match

well, and the average error between the simulated value and the experimental value is within 5%, which belongs to the reasonable error range. Figure 13a–d is the comparison between the simulated and experimental values of the main emissions of the engine under the four experimental conditions. From Figure 13, the experimental values of $NO_X$, HC and CO emissions are close to the simulated values. The error rates of $NO_X$ and HC emissions are about 14%, and the error rate of CO emissions is about 20%. The emissions values of each of the emissions are in an order of magnitude. In summary, the constructed diesel/hydrogen dual fuel combustion mechanism can better predict the combustion and emissions characteristics of diesel/hydrogen dual fuel engines.

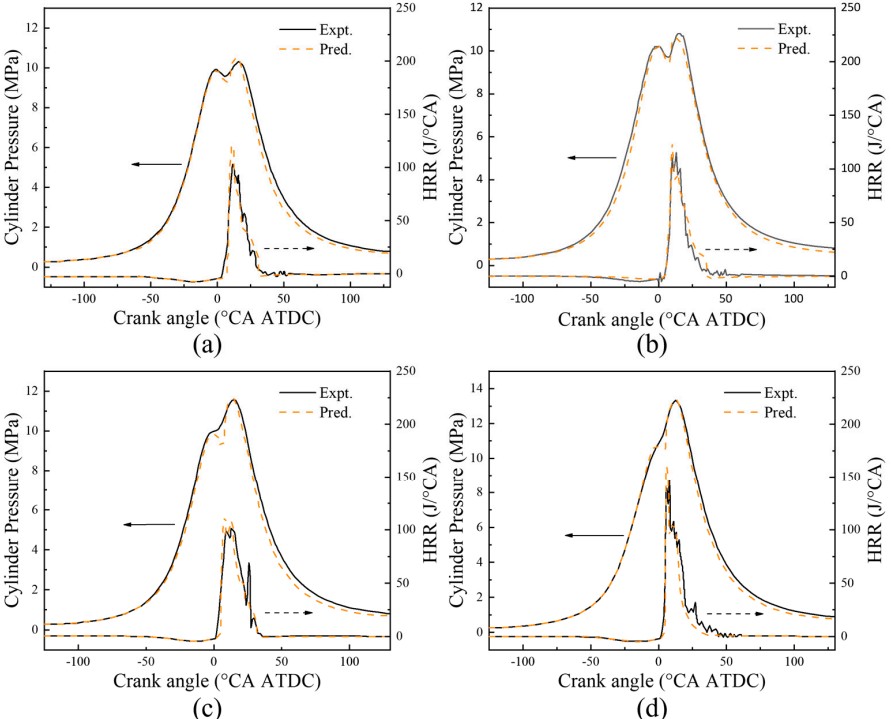

**Figure 12.** Comparison of simulated values and operation values of cylinder pressure and heat release rate of engine under different operation conditions: (**a**) Engine speed = 1800 r/min, Torque = 322 N·m, Hydrogen replacement rate = 0%; (**b**) Engine speed = 1800 r/min, Torque = 322 N·m, Hydrogen replacement rate = 30%; (**c**) Engine speed = 1800 r/min, Torque = 365 N·m, Hydrogen replacement rate = 0%; (**d**) Engine speed = 1800 r/min, Torque = 365 N·m, Hydrogen replacement rate = 27%.

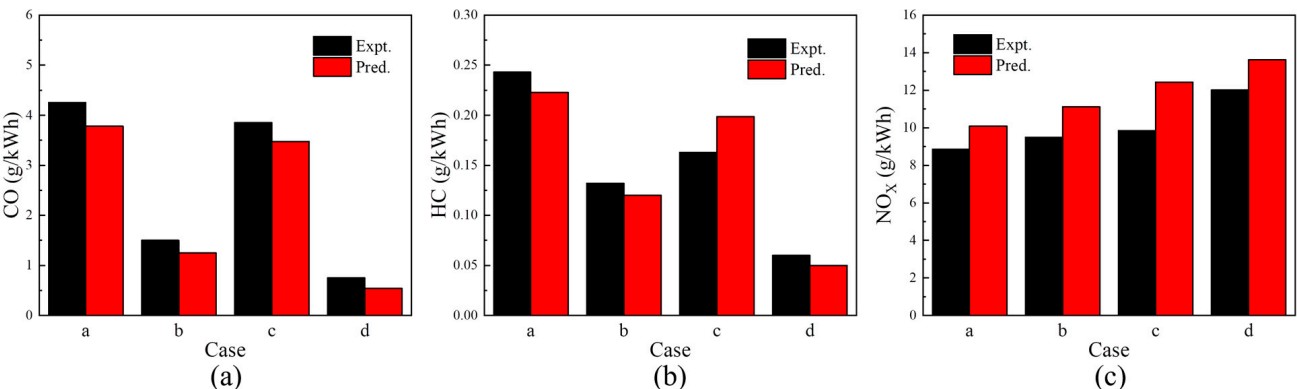

**Figure 13.** Comparison of simulated and operation values of engine pollution emissions under different operation conditions: (**a**) CO emissions under 4 operation conditions; (**b**) HC emissions under 4 operation conditions; (**c**) $NO_X$ emissions under 4 operation conditions.

The operation condition (b) in Table 6 is selected for numerical simulation. According to the proportion of pilot injection quantity and the range of starting angle commonly used in previous academic research [57–66], in this paper, the pilot injection strategies mainly select five groups of pilot mass percent and five groups of pilot injection timing for calculation. As shown in Table 7, the pilot mass percent is defined as the percent of the pilot injection quantity to the total circulating oil quantity, and the sum of the pilot injection quantity and the main injection quantity is 100%.

**Table 7.** Simulation research scheme.

| Parameter | Value |
| --- | --- |
| Rated speed (r/min) | 1800 |
| Torque (N·m) | 322 |
| Hydrogen replacement rate/% | 30% |
| Total injection diesel (mg) | 37.8 |
| Pilot mass percent/% | 0%, 5%, 10%, 15%, 20% |
| Pilot injection timing (CA BTDC) | 10°, 20°, 30°, 40°, 50° |
| Main injection timing (CA BTDC) | 0°, 2°, 4°, 6°, 8° |

## 5. Results and Discussions

### 5.1. Effect of Pilot Injection Strategies on Combustion and Emissions Characteristics of Diesel/Hydrogen Dual Fuel Engine

5.1.1. Combustion Characteristics

Figure 14 shows the effects of different pilot injection strategies on combustion characteristics. The corresponding crankshaft angle when the heat release reaches 10% is defined as the combustion starting point CA10; the corresponding crankshaft angle when the heat release reaches 50% is defined as the combustion center of mass CA50.

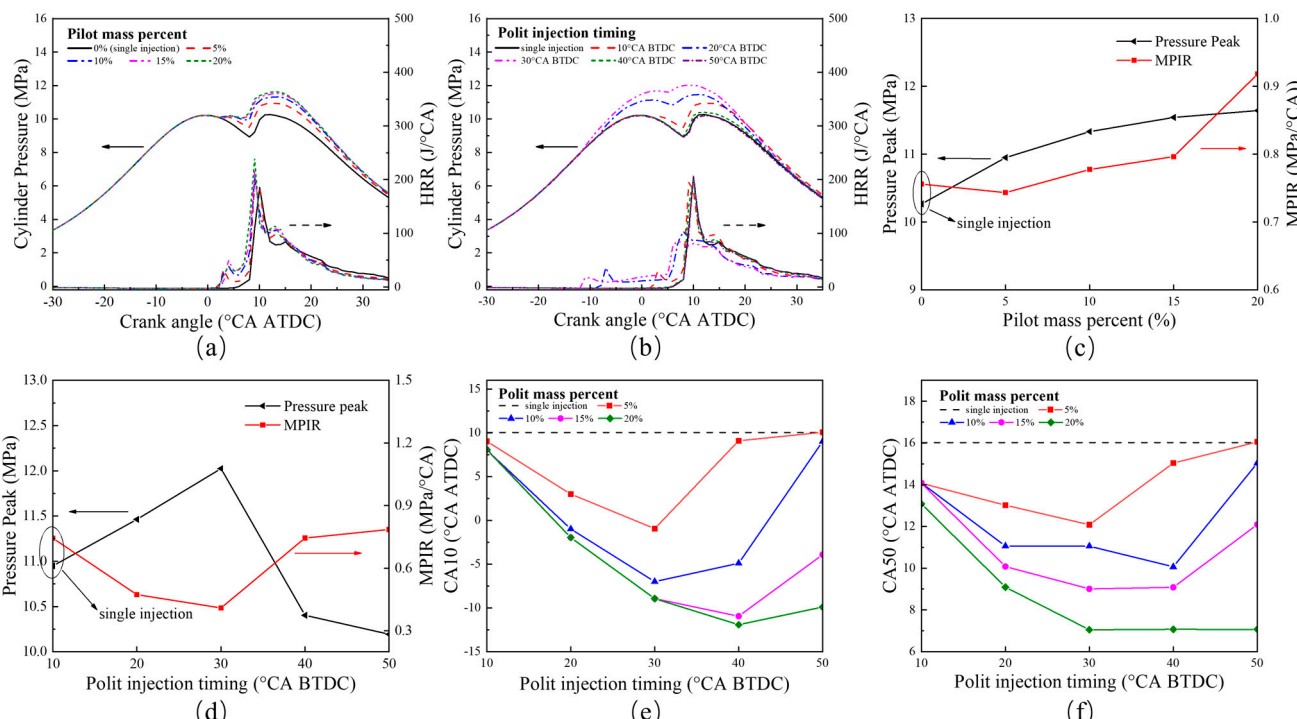

**Figure 14.** Effects of different pilot injection strategies on combustion characteristics: (**a**) Cylinder pressure and HRR under different pilot mass percentages; (**b**) Cylinder pressure and HRR under different pilot injection timings; (**c**) Pressure peak and MPIR under different pilot mass percentages; (**d**) Pressure peak and MPIR under different pilot injection timings; (**e**) CA10 under different pilot mass percentages and pilot injection timings; (**f**) CA50 under different pilot mass percentages and pilot injection timings.

As shown in Figure 14a,c,e,f, the pilot injection timing is fixed at 10° CA BTDC, and the main injection timing is fixed at 4° CA BTDC. As the pilot mass percent increases from 0% to 20%, the peak cylinder pressure and MPIR show an upward trend, the corresponding phases of CA10 and CA50 are advanced, and the peak heat rate caused by the pilot injection amount shows an upward trend. This is mainly because under the condition of early pilot injection timing, with the increase in pilot mass percent, more pilot injection diesel is mixed with the hydrogen–air mixture in the compression stage, and the temperature in the cylinder increases faster, which promotes the atomization and evaporation of the main injection, resulting in the increase in ignition area and combustion rate. More pre-mixed gas is consumed before the main injection diesel, and more heat is released in the pilot injection stage. Therefore, the MPIR and peak heat release rate in the cylinder increase. Therefore, CA10 is advanced and CA50 is closer to the TDC.

Figure 14 also characterizes the effect of pilot injection timing on the combustion characteristics of a diesel/hydrogen dual fuel engine. From Figure 14b,d,e,f, the pilot mass percent is fixed at 5%, and the main injection timing is fixed at 4° CA BTDC. With the advance of the pilot injection timing, the peak cylinder pressure first increases and then decreases, the peak MPIR and the peak heat release rate first decrease and then increase, and CA10 and CA50 first advance and then delay. This is mainly due to the advance of the pilot injection timing (10~30° AC BTDC), the injection interval between the pilot injection and the main injection is close, and the advance of CA10 and CA50 makes more mixtures participate in the combustion before the TDC, which makes the peak cylinder pressure increase, and the shortening of the ignition delay period causes the diesel/hydrogen mixture to decrease, so the peak heat release rate and MPIR decrease; with the advance of pilot injection timing (30~50° AC BTDC), the injection interval of pilot injection and main injection is far, and the influence of pilot injection on the main injection is gradually weakened. The CA10 and CA50 of the mixture are continuously delayed and away from the TDC, resulting in more fuel participating in combustion after the TDC, resulting in a decrease in in-cylinder pressure. The growth of the ignition delay period provides sufficient time for the mixing of pilot injection diesel and hydrogen mixture, and more ignition cores are distributed in the hydrogen mixture. The combustion rate of the mixture accelerates, which increases the peak heat release rate and MPIR.

5.1.2. Emission Characteristics

Figure 15 shows the changes in $NO_X$ and CO emissions under different pilot injection strategies; Figure 16 shows the mass concentration distribution of $NO_X$ and CO under different pilot injection strategies CA50. The generation of $NO_X$ is related to the in-cylinder temperature, oxygen concentration and reaction time, and CO is mainly derived from the incomplete combustion of diesel.

The effect of pilot injection quantity on $NO_X$ and CO emissions is shown in Figure 15a. It can be seen from Figure 15 a that when the pilot injection timing is fixed, with the increase in the pilot mass percent, the CO emissions gradually decrease and the $NO_X$ emissions gradually increase. This is because with the increase in pilot injection amount, the ignition source of the mixture in the cylinder is more widely distributed, which promotes the combustion rate of the mixture, and the overall combustion is more complete, resulting in higher combustion pressure and temperature. Therefore, CO emissions are reduced, and $NO_X$ emissions are increased. As shown in Figure 16a, CO emissions are the highest in single injection (pilot mass percent is 0%), and they are mainly concentrated in the center of piston compression clearance. With the increase in pilot mass percent, the concentration at the center of piston compression clearance is lower and lower. When the pilot mass percent is 20%, CO is mainly distributed in the pit of the combustion chamber and the edge of piston compression clearance. The $NO_X$ emissions is the lowest in a single injection (pilot mass percent is 0%), and it is sparsely distributed in the combustion chamber. With the increase in the pilot mass percent, the concentration of $NO_X$ in the center of the piston compression clearance is higher and higher.

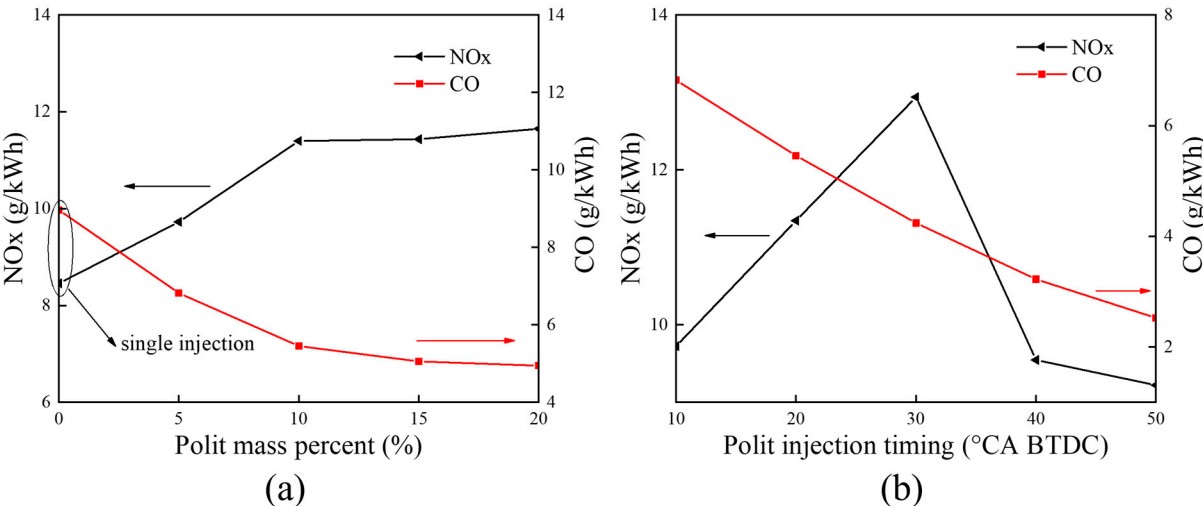

**Figure 15.** Changes of NO$_X$ and CO emissions under different pilot injection strategies: (**a**) NO$_X$ and CO emissions under different pilot mass percentages; (**b**) NO$_X$ and CO emissions under different pilot injection timings.

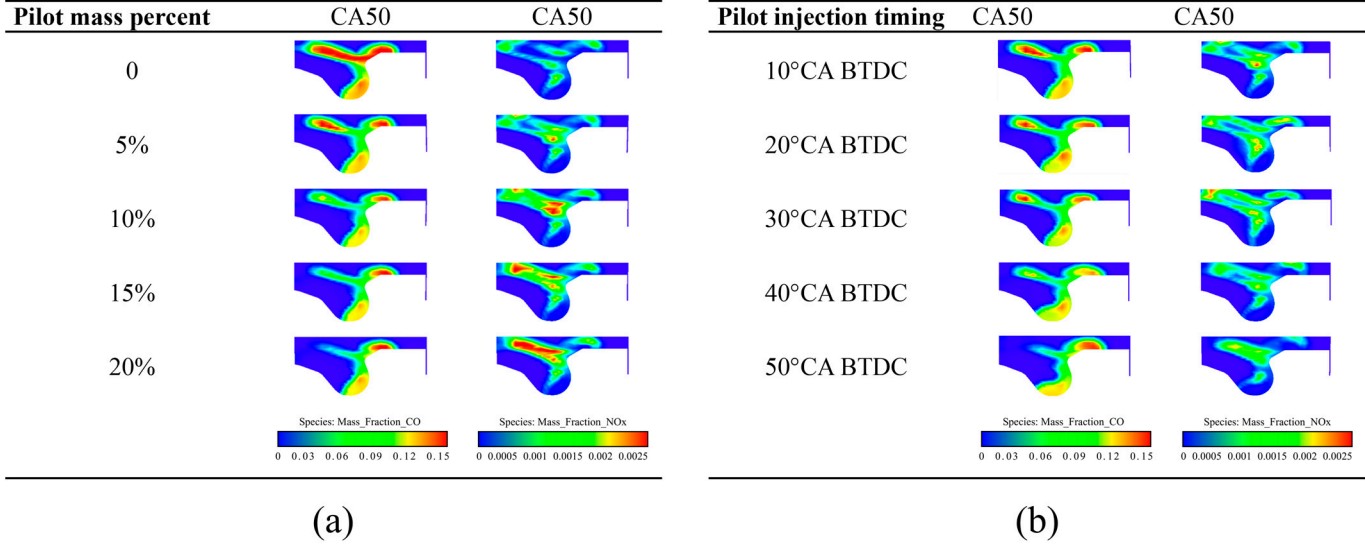

**Figure 16.** The mass concentration distribution of NO$_X$ and CO under different pilot injection strategies: (**a**) Mass concentration distribution of NO$_X$ and CO under different pilot mass percentages; (**b**) Mass concentration distribution of NO$_X$ and CO under different pilot injection timings.

The effect of pilot injection timing on NO$_X$ and CO emissions is shown in Figure 15b. From Figure 15b, when the pilot mass percent is fixed, the CO emissions decrease gradually with the advance of the pilot injection timing, and the NO$_X$ emissions increase first and then decrease. With the advance of pilot injection timing (10~30° CA BTDC), the advance of CA10 and CA50 makes more hydrogen mixture participate in combustion before TDC, and the combustion rate is accelerated, resulting in higher in-cylinder combustion pressure and temperature. Therefore, CO emissions are reduced, and NO$_X$ emissions are increased. With the advance of the pilot injection timing (30~50° CA BTDC), CA10 and CA50 are gradually delayed and far away from the TDC. Most of the hydrogen mixture combustion occurs in the expansion stroke, resulting in the reduction in combustion pressure and temperature in the cylinder. At the same time, the longer mixing time makes the mixture distribution more uniform, which inhibits the generation of NO$_X$. The uniform mixture distribution at the earlier pilot injection timing means a wider ignition source, shortens the distance from the flame to the edge mixture, and promotes the combustion of the hydrogen

mixture, so the CO emissions is reduced. As shown in Figure 16b, with the advance of pilot injection timing (10~50° CA BTDC), the mass concentration of CO gradually decreases, and the distribution area gradually shifts from the piston compression clearance to the piston bottom and the clearance edge area. With the advance of pilot injection timing (10~30° CA BTDC), $NO_X$ gradually becomes concentrated, and it is mainly distributed in the central area of piston compression clearance. When the pilot injection timing is further advanced from 30° CA BTDC to 50° CA BTDC, the $NO_X$ mass concentration gradually decreases, and the distribution area is mainly in the center area of the piston clearance.

### 5.2. Effect of Main Injection Timing on Combustion and Emissions Characteristics of Diesel/Hydrogen Dual Fuel Engine

#### 5.2.1. Combustion Characteristics

Figure 17 shows the effect of different main injection timings on combustion characteristics. As shown in Figure 17, when the fixed pilot mass percent is 5% and the pilot injection timing is 10° CA BTDC, with the advance of the main injection timing, the peak heat release rate and MPIR first decrease slightly and then increase, and the peak cylinder pressure increases. The corresponding phase of the peak heat release rate and the peak cylinder pressure moves forward as a whole. When the main injection timing is advanced to 6° CA BTDC, the MPIR is 1.3 MPa/° CA and exceeds the diesel engine MPIR limit of 1.2 MPa/° CA. This is due to the shortening of the ignition delay period with the advance of the main injection timing (0~2° CA BTDC), resulting in a decrease in the diesel/hydrogen mixture as well as a decrease in the peak heat release rate and MPIR. When the main injection timing is further advanced from 2° CA BTDC to 8° CA BTDC, due to the close interval between pilot injection and main injection, the advance of combustion phase makes the fuel burn faster and release more heat at the end of the compression stroke. Therefore, the peak heat release rate, peak cylinder pressure and MPIR all increase.

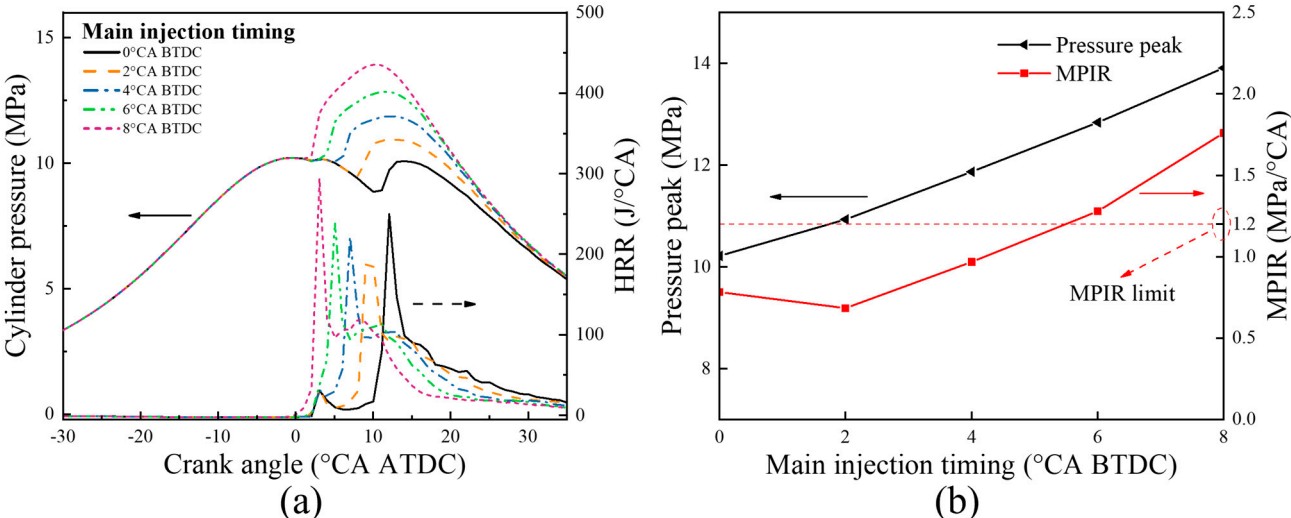

**Figure 17.** Effects of different main injection timings on combustion characteristics: (**a**) Cylinder pressure and HRR under different main injection timings; (**b**) Pressure peak and MPIR under different main injection timings.

#### 5.2.2. Emission Characteristics

Figure 18 shows changes of $NO_X$ and CO emissions under different main injection timings. Figure 19 shows the mass concentration distribution of $NO_X$ and CO under different main injection timing CA50.

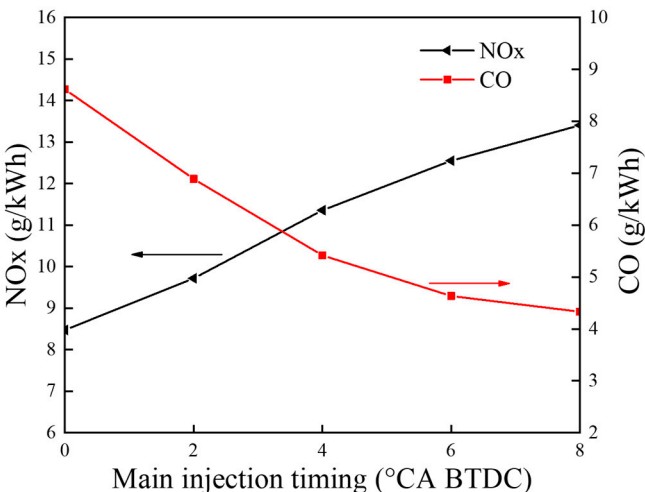

**Figure 18.** Changes of NO$_X$ and CO emissions under different main injection timings.

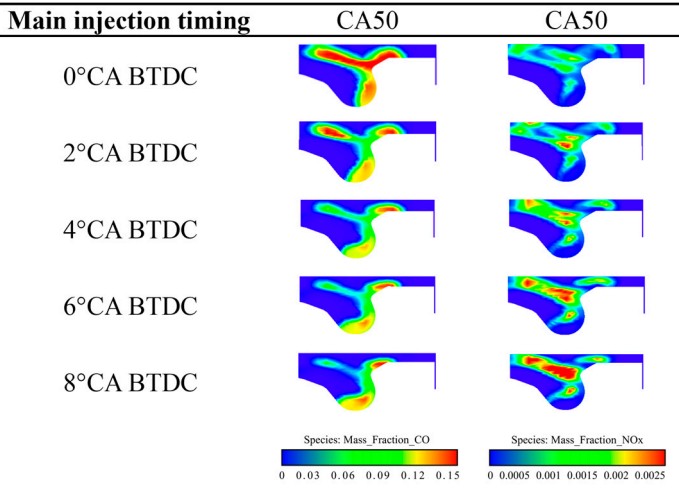

**Figure 19.** The mass concentration distribution of NO$_X$ and CO under different main injection timings.

The effect of main injection timings on NO$_X$ and CO emissions is shown in Figure 18. With the advance of main injection timing, CO emissions gradually decrease, and NO$_X$ emissions gradually increase. This is because with the advance of the main injection timing, the interval between the pilot injection and the main injection is close, and the advance of the combustion phase enables the in-cylinder mixture to burn and release heat faster, prompting the in-cylinder temperature and pressure to continue to rise. Therefore, CO emissions decrease and NO$_X$ emissions increase. As shown in Figure 19, CO emissions are highest at the main injection timing of 0° CA BTDC and are mainly concentrated in the central area of the piston compression clearance. With the advance of the main injection timing, the concentration of CO in the central area of the piston compression clearance is becoming lower and lower, and it gradually concentrates at the bottom of the combustion chamber pit. For NO$_X$ emissions, it is mainly distributed in the center area of piston compression clearance, and with the advance of main injection timing, the distribution position of NO$_X$ does not change roughly, while the red area with a high concentration of NO$_X$ is becoming higher and higher. Therefore, to reduce the NO$_X$ emissions of the diesel/hydrogen dual fuel engine, the main injection timing can be appropriately delayed.

## 6. Conclusions

In this study, a reduced diesel/hydrogen dual fuel combustion mechanism was proposed. Based on the dual fuel combustion mechanism, a three-dimensional CFD model was

coupled to study the effects of diesel pilot injection quantity and pilot injection timing and main injection timing on the combustion and emission characteristics of a diesel/hydrogen dual fuel engine. The main conclusions are as follows:

(1) 70% n-decane and 30% $\alpha$-methylnaphthalene (IDEA reference fuel) were selected as diesel substitutes. The sub-mechanisms of n-decane, $\alpha$-methylnaphthalene, $NO_X$, PAH, soot and $H_2/C_1$-$C_3$ were combined to obtain the diesel/hydrogen dual fuel combustion mechanism. A reduced diesel/hydrogen dual fuel combustion mechanism with 191 components and 847 elementary reactions was finally obtained by optimization.

(2) The combustion mechanism of diesel/hydrogen dual fuel was verified by ignition delay, JSR and laminar flame speed. In the verification of the ignition delay time of n-decane/$\alpha$-methylnaphthalene blends and IDEA reference fuel for diesel and hydrogen, the average error is within an order of magnitude; in the JSR oxidation verification of diesel and hydrogen by IDEA reference fuel, the average error is less than 6%. In the laminar flame verification of diesel and hydrogen with n-decane, $\alpha$-methylnaphthalene and IDEA reference fuel, the average error is less than 16%.

(3) The reduced diesel/hydrogen dual fuel mechanism is coupled with the three-dimensional CFD model to verify the combustion and emission characteristics of the dual fuel engine under four operation conditions. The results show that the variation trend of in-cylinder pressure and combustion heat release rate is consistent with the simulated value, and the average error is less than 5%. The variation trend of $NO_X$, CO and HC emissions is consistent with the simulated values, and the average error is less than 17%.

(4) With the increase in the pilot injection quantity, the peak heat release rate, the peak cylinder pressure and MPIR show an upward trend, the corresponding phase of CA10 and CA50 is advanced, the CO emissions is reduced, and the $NO_X$ emission is increased. With the advance of pilot injection timing, the peak value of cylinder pressure increases first and then decreases, the peak value of MPIR and heat release rate decreases first and then increases, the corresponding phases of CA10 and CA50 advance first and then delay, CO emissions decrease gradually, and $NO_X$ emissions increase first and then decrease. With the advance of the main injection timing, the peak heat release rate and MPIR first decreased slightly and then increased, the peak cylinder pressure increased, the corresponding phase of the peak heat release rate and the peak cylinder pressure moved forward as a whole, the CO emission decreased, and the $NO_X$ emission increased.

**Author Contributions:** Conceptualization, S.L. and H.D.; methodology, S.L.; software, L.X.; validation, S.L., H.D. and L.S.; formal analysis, L.X.; investigation, L.X.; resources, S.L., L.S. and L.X.; data curation, L.X.; writing—original draft preparation, L.X.; writing—review and editing, S.L.; visualization, S.L.; supervision, Y.B.; project administration, S.L. and L.X.; funding acquisition, S.L. and L.X. All authors have read and agreed to the published version of the manuscript.

**Funding:** This research was funded by the National Natural Science Foundation of China under grant 52066008, study on the influence of fuel design coupling fuel injection control on combustion characteristics of diesel engine under grant 2021J0057.

**Data Availability Statement:** The data used to support the findings of this study are included within the article.

**Conflicts of Interest:** The authors declare no conflict of interest.

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
