# Peer review of "Study on the Combustion Mechanism of Diesel/Hydrogen Dual Fuel and the Influence of Pilot Injection and Main Injection"

_processes, doi:10.3390/pr11072122_

Round 1
Reviewer 1 Report
This paper introduced a study on the combustion mechanism of diesel / hydrogen dual fuel and the influence of pilot injection and main injection. some concern should be addressed:
1, there are some error between the experimental and predicted ignition delays for α-methylnaphthalene/n-decane mixtures, IDEA reference fuel and hydrogen, the author should explain in detail.
2, why the author choose such late injection include pilot injection and main injection? the peak pressure located 10-20 ° CA ATDC.
3, Figure 17(a) are wrong, the compression peak is not located in 0° CA.
this paper should be edited by English native speaker.
Reviewer 2 Report
The paper titled "Study on the combustion mechanism of diesel / hydrogen dual fuel and the influence of pilot injection and main injection" presents a comprehensive investigation into the effects of diesel pilot injection quantity, pilot injection timing, and main injection timing on the combustion and emission characteristics of a diesel/hydrogen dual fuel engine. The authors propose a reduced combustion mechanism and couple it with a three-dimensional CFD model to simulate and validate the engine performance under different operating conditions.
The first major contribution of this study is the development of a reduced diesel/hydrogen dual fuel combustion mechanism. The authors combined sub-mechanisms of n-decane, α-methylnaphthalene, NOX, PAH, soot, and H2/C1-C3 to construct a comprehensive mechanism consisting of 191 components and 847 elementary reactions. This optimization process ensures a reliable and accurate representation of the dual fuel combustion process.
To validate the proposed mechanism, the authors performed ignition delay, JSR, and laminar flame speed tests. The results demonstrate good agreement between experimental and simulated values, with average errors within acceptable limits. This verification process ensures the reliability and accuracy of the combustion mechanism for subsequent simulations.
The authors then coupled the reduced combustion mechanism with a three-dimensional CFD model to simulate the combustion and emission characteristics of the dual fuel engine under various operating conditions. The simulated results were compared with experimental data, and the agreement was found to be satisfactory. The in-cylinder pressure, combustion heat release rate, and emissions of NOX, CO, and HC followed similar trends to the simulated values, with average errors below 17%.
Furthermore, the authors investigated the effects of diesel pilot injection quantity, pilot injection timing, and main injection timing on engine performance. Increasing the pilot injection quantity resulted in higher peak heat release rate, peak cylinder pressure, and maximum pressure rise rate (MPIR). The corresponding combustion angles (CA10 and CA50) were advanced, CO emissions were reduced, and NOX emissions increased. Advancing the pilot injection timing led to an initial increase and subsequent decrease in peak cylinder pressure and MPIR, while CA10 and CA50 advanced and then delayed. CO emissions decreased gradually, whereas NOX emissions initially increased and then decreased. Advancing the main injection timing resulted in a slight decrease followed by an increase in peak heat release rate and MPIR. Peak cylinder pressure increased, and the corresponding combustion angles moved forward as a whole. CO emissions decreased, while NOX emissions increased.
Overall, this paper provides a valuable contribution to the understanding of diesel/hydrogen dual fuel combustion and its impact on engine performance and emissions. The developed reduced combustion mechanism and the coupled CFD model demonstrate good accuracy and reliability in predicting the combustion and emission characteristics. The findings regarding the effects of injection quantity and timing on engine performance are insightful and can contribute to the optimization of dual fuel engine operation. However, it is worth noting that further experimental validation of the simulated results would strengthen the reliability and significance of the study.
The manuscript is well-presented in all its sections, and only minor modifications are recommended before publication:
- In the abstract, there are some errors with capital letters after semicolons; please correct them.
- In the numerical CFD model section, please specify whether grid convergence was performed or if only comparisons with experiments were made. If the former, please explain how the grid size was determined.
- In the same CFD section, please specify the values of Y+ and Re.
